# Extracellular Vesicle-Packaged miR-195-5p Sensitizes Melanoma to Targeted Therapy with Kinase Inhibitors

**DOI:** 10.3390/cells12091317

**Published:** 2023-05-05

**Authors:** Nathalia L. Santos, Silvina O. Bustos, Patricia P. Reis, Roger Chammas, Luciana N. S. Andrade

**Affiliations:** 1Center for Translational Research in Oncology (LIM24), Instituto do Câncer do Estado de São Paulo, Hospital das Clínicas da Faculdade de Medicina da Universidade de São Paulo, Comprehensive Center for Precision Oncology, Universidade de São Paulo, São Paulo 01246-000, Brazil; nathalialeal@usp.br (N.L.S.); silvina.bustos@hc.fm.usp.br (S.O.B.); 2Department of Surgery and Orthopedics and Experimental Research Unity (UNIPEX), Faculdade de Medicina, Universidade Estadual Paulista (UNESP), Botucatu 18618-687, Brazil; patricia.reis@unesp.br

**Keywords:** melanoma, miR-195-5p, extracellular vesicles, MAPK inhibitors

## Abstract

Management of advanced melanoma remains challenging, with most BRAF (B-Raf proto-oncogene, serine/threonine kinase)-mutated metastatic patients relapsing within a few months upon MAPK inhibitors treatment. Modulation of tumor-derived extracellular vesicle (EVs) cargo with enrichment of antitumoral molecules is a promising strategy to impair tumor progression and increase treatment response. Herein, we report that restored expression of miR-195-5p, down-regulated in melanoma favoring drug resistance, increases the release of EVs enriched in the tumor suppressor miRNAs, miR-195-5p, miR-152-3p, and miR-202-3p. Incorporating these EVs by bystander tumor cells resulted in decreased proliferation and viability, accompanied by a reduction in CCND1 and YAP1 mRNA levels. Upon treatment with MAPK inhibitors, miR-195 EVs significantly decreased BCL2-L1 protein levels and increased cell death ratio and treatment efficacy. Additionally, EVs exogenously loaded with miR-195-5p by electroporation reduced tumor volume in vivo and impaired engraftment and growth of xenografts implanted with melanoma cells exposed to MAPK inhibitors. Our study shows that miR-195-5p antitumoral activity can be spread to bystander cells through EVs, improving melanoma response to targeted therapy and revealing a promising EV-based strategy to increase clinical response in patients harboring BRAF mutations.

## 1. Introduction

Melanoma is the most aggressive form of skin cancer and is considered the major skin cancer-related cause of death. Although recent breakthroughs have significantly changed clinical outcomes for advanced-stage, unresectable, or metastatic patients, most of them acquire or present intrinsic resistance to standard-of-care approaches, leading to a poor prognosis. Combined targeted therapy, with BRAF (B-Raf proto-oncogene, serine/threonine kinase) and MEK (mitogen-activated protein kinase kinase) inhibitors (MAPKi), is considered the standard first-line treatment for BRAF-mutated patients. BRAF mutation is the most frequent genetic alteration in melanomas, comprising around 50% of the cases. Substitution of valine with glutamate in codon 600 (V600E) is the most common mutation in the BRAF gene and leads to the constitutive activation of the mitogen-activated protein kinase (MAPK) pathway through subsequent phosphorylation of the kinases MEK and ERK, leading to the constitutive transcription of genes related to cell proliferation and drug resistance. Despite the improvements in patient outcomes achieved by simultaneously targeting BRAF and MEK, tumor resistance and recurrence remain challenging, resulting in a 5-year overall survival of around 20% [1,2].

Over the past years, extracellular vesicle (EVs)-mediated transfer of information became one of the critical mechanisms regulating tumor treatment response. EVs are nanometric particles enclosed in a lipid bilayer released from cells as mediators of intercellular communication. They are composed of distinct populations, mainly including exosomes (small vesicles, 30–150 nm) derived from the endosomal pathway and microvesicles (large vesicles, 0.1–1 µm) derived from the budding of the plasma membrane [3]. Cancer-derived EVs have been shown to favor resistant phenotypes by transferring pro-tumoral molecules, including nucleic acids, proteins, and metabolites [4,5,6,7].

Particularly, EV miRNA cargo has been associated with melanoma treatment response and patient outcome. MiRNAs typically regulate gene expression by binding to the 3’ untranslated region (UTR) of mRNAs, repressing translation. The presence of tumor-suppressive miRNAs within EVs is related to increased response rates and better prognosis [8,9], whereas lower levels of these molecules correlate with greater melanoma aggressiveness and treatment relapse [10,11]. MiR-195-5p is a miRNA that controls melanoma therapy response. As previously shown by our group, miR-195-5p expression is down-regulated along melanoma development, favoring cell proliferation and drug resistance. Restoring its expression exerts an anti-proliferative effect, increasing BRAF-mutated cancer cells’ sensitivity to alkylating chemotherapy and vemurafenib-targeted therapy [12].

Considering that the content and abundance of molecules within EVs are associated with the phenotypic state of their parental cells, we aimed to analyze whether restoring miR-195-5p expression in BRAF-mutated melanoma cells could modulate the miRNA profile within EVs and the role of these particles in bystander recipient cells response to combined targeted treatment. Upon miR-195-5p overexpression, melanoma cells release EVs enriched with tumor suppressors miRNAs (miR-195-5p, miR-152-3p, and miR-202-3p), which impaired the growth of bystander cells after EVs uptake. At the molecular level, this effect was accompanied by a reduction in CCND1 and YAP1 mRNA levels. Additionally, the combined treatment with EVs and MAPKi sensitized cells to therapy in vitro, decreasing BCL2-L1 protein levels. Finally, to further explore the translational potential of miR-195-5p delivery by EVs, we exogenously loaded this miRNA into melanoma-derived vesicles for in vivo tumor treatment. These EVs significantly reduced tumor growth kinetics and repopulation capacity of melanoma cells exposed to MAPKi, indicating that EVs-based delivery of miR-195 can be a promising strategy to improve therapy response in melanoma to MAPK inhibition.

## 2. Materials and Methods

### 2.1. Cell Culture

BRAF-mutated (BRAFV600E) human melanoma cell lines A375, SKMel-5, and SKMel-28 were cultured at 37 °C and 5% CO_2_, using MEM supplemented with 10% fetal bovine serum (FBS) and sodium pyruvate (1 mM) for SKMel-5 and SKMel-28, and DMEM added of 10% FBS for A375. A375 cells were purchased from American Type Culture Collection (ATCC, VA, USA, CRL-161), while SK-MEL-5 and SKMel-28 were donated by Lloyd J. Old from Ludwig Institute for Cancer Research and Memorial Sloan Kettering Cancer Center. EVs-depleted FBS was used for EVs experiments (centrifuged for 2 h, 4 °C, 100,000× *g*). Cells were routinely screened for Mycoplasma contamination. Our experiments were conducted using these three cell lines to avoid a lack of representation of melanoma heterogeneity.

### 2.2. Mimics Transfection

Cells were transfected with 10 nM of double-stranded miRNA Mimic Syn-hsa-miR-195-5p (5′-UAGCAGCACAGAAAUAUUGGC-3′) (QIAGEN, catalog number MSY0000461, Hilden, Germany) or ALL STARS Negative control siRNA (QIAGEN, catalog number 1027292) using Lipofectamine RNAiMAX transfection reagent (Thermo Fischer Scientific, Waltham, MA, USA, catalog number 13778075). MiRNA mimics were reversed transcribed according to the manufacturer’s instructions. First, the miRNA/Lipofectamine mixes were prepared and incubated for 20 min at room temperature. Next, miRNA/Lipofectamine complexes were added to 6-well plates, followed by cells (10^5^ cells/well) suspended in their respective medium.

### 2.3. EVs Isolation and Characterization

EVs were isolated by differential ultracentrifugation, as described by Théry et al. [13]. Seventy-two hours after transfection, conditioned media were collected and submitted to centrifugation at 300× *g* for 10 min followed by 2000× *g* for 10 min, both at room temperature. Next, the supernatants were ultracentrifuged at 10,000× *g* for 30 min, followed by 100,000× *g* for 2 h at 4 °C (Beckman Coulter Optima XE-90, rotor SW28, CA, USA). EV pellets were resuspended in phosphate-buffered saline (PBS) according to total cell number and stored at −80 °C. EV characterization was conducted according to the 2018/2022 consensus guidelines on Minimal Information for Studies of Extracellular Vesicles [14]. Nanoparticle Tracking Analyses (NTA) determined EVs’ concentration and size using NanoSight NS300 (Malvern, UK), with 5 measurements of 60 s each, at 25 °C. Western blot and transmission electron microscopy analyzed classical membrane markers and EV structure, respectively.

#### Transmission Electron Microscopy (TEM)

EVs were fixed in a 2% paraformaldehyde, 2% glutaraldehyde solution (1:1) in sodium cacodylate buffer (pH 7.4) for 30 min at 4 °C, followed by washing with PBS and ultracentrifugation at 100,000× *g*. Next, EVs (10 µL) were mounted in TEM grids, contrasted with uranyl acetate 2% for 15 min, and washed in distilled water. Fluid excess was blotted in filter paper, air-dried, and examined using a ZEISS Leo 906 electron microscope (Carl Zeiss, Jena, Germany).

### 2.4. RNA Extraction and RT-qPCR

Total RNA was extracted from cells using Trizol reagent (Thermo Fischer Scientific). miRCURY^TM^ RNA Isolation Kit (EXIQON-QIAGEN) or miRNeasy Tissue/Cells Advanced Mini Kit (QIAGEN, catalog number 217684) was used for EVs RNA isolation. Previous to RNA extraction, EVs were treated with Proteinase K (1 mg/mL) for 10 min followed by RNAse A (400 ng/µL; Sigma-Aldrich, MA, USA) for an additional 30 min at 37 °C, according to VERWEIJ et al. [15]. For miR-195-5p expression analysis, cDNA synthesis and quantitative RT-PCR were carried out using TaqMan Small RNA assays (Thermo Fischer Scientific, catalog number 4398987). RNU48 was used as endogenous control. High-Capacity Reverse Transcription Kit and Power SYBR Green PCR master mix (Thermo Fischer Scientific, catalog number 4309155) was used for cDNA synthesis and qPCR reactions. GAPDH, β-ACTIN, and HPRT were used as endogenous control (Appendix A). Data were analyzed using the 2^−ΔΔCt^ method.

### 2.5. Gene Expression Profile

According to the manufacturer’s instructions, the gene expression profile was determined by microfluidic RT-qPCR using Juno-Biomark HD equipment (Fluidigm, San Francisco, CA, USA). Then, 35 ng of total RNA was converted in cDNA, followed by pre-amplification. β-ACTIN, B2M, GUSB, HPRT, RPLP0, and TFRC were used as endogenous control (Appendix A). NormFinder software (version 5) was used to determine the most stable endogenous genes within the experimental groups. Data were analyzed by 2^−∆∆Ct^. R Studio software was used for sample clusterization in a heatmap using the pheatmap package.

### 2.6. Western Blotting (WB)

Proteins were obtained using the RIPA buffer (Sigma) containing protease and phosphatase inhibitors and incubated for 15 min at 4 °C. For EVs, samples were sonicated using a probe sonicator (Fisher Brand) for 3 pulses of 5 s each at low amplitude. Then, 30–50 µg of cellular protein and protein from 10^10^ EVs were loaded onto 10% SDS-PAGE (0.375 M Tris, pH 8.8, 0.1% SDS, 10% acrylamide, 0.03% ammonium persulfate (APS), and 0.06% N, N, N′, N′-tetramethyl ethylenediamine (TEMED)). PVDF membranes were blocked with 5% BSA in 0.1% TBS-Tween for 1 h at room temperature, followed by incubation with primary antibodies overnight at 4 °C and secondary antibodies (Appendix A) for 1 h at room temperature. Protein bands were visualized with the chemiluminescent substrate ECL (GE Healthcare, Chicago, IL, USA).

### 2.7. miRNA Profile

Analysis of cellular and EVs miRNA content was conducted using nCounter Human miRNA Assay (Nanostring Technologies, Seattle, WA, USA), containing 799 probes for detecting human mature miRNAs based on miRbase version 17, accordingly to the manufacturer’s instructions. Total RNA from cells (100 ng) and vesicles (~20 ng) was extracted using miRNeasy Tissue/Cells Advanced Mini Kit (QIAGEN). Raw miRNA count was analyzed using the NanoString technologies nSolver analysis software v3.0. The highest count among the negative control probes in each group of samples (cells and EVs) was used as a detection threshold to avoid false-positive data. Data were normalized using the most stable miRNAs among the groups as endogenous controls.

### 2.8. EV Incorporation Assay

EVs were previously stained with PKH26 (Sigma-Aldrich) according to instructions. First, 2 µL of PKH26 dye was diluted in 500 µL of Diluent C and mixed with 500 µL of EVs (1:1). The mixtures were incubated for 5 min at room temperature. Excess dye was removed by filtration using MW3000 columns (Invitrogen—Thermo Fischer Scientific). Stained EVs were added at different concentrations to the cell media 24 h after plating. After overnight incubation, cells were washed with PBS to remove non-incorporated EVs. Images were obtained using EVOS Fluorescence Microscopy (Thermo Fisher Scientific).

### 2.9. Functional Assays

#### 2.9.1. Cell Viability Assay

Cell viability was evaluated by yellow tetrazolium salt (MTT) colorimetric assay. Cells were previously seeded in 96-well plates (10^3^ cells/well). Next, different concentrations of EVs were added daily. After 72 h in the presence of EVs, 10 µL of the MTT solution (5 mg/mL) was added, followed by incubation for 2 h at 37 °C (5% CO_2_). Cells were lysed with dimethylsulfoxide (DMSO). Light absorbance at 570 nm was measured in a spectrophotometer.

#### 2.9.2. Cell Proliferation Assay

Cells were seeded in 24-well plates (10^4^ cells/well). The following day, EVs were added daily for 72 h at a concentration of 10^8^ particles/mL for A375 and SKMel-28 and 10^7^ particles/mL for SKMel-5. The number of viable and non-viable cells was determined using a Neubauer chamber and trypan blue solution (0.1%).

#### 2.9.3. Cell Cycle Profile

Cell cycle analysis was performed by propidium iodide (PI) staining. Cells were detached from plates with trypsin, fixed in 70% ethanol, and kept at −20 °C for at least 2 h. Next, samples were washed with PBS and stained with PI solution (20 µg/mL propidium iodide, 5 µg/mL RNase A, 1% Triton X-100) for 30 min protected from light. Data were acquired using Attune cytometer (Thermo Fisher Scientific). Cell cycle distribution (sub-G1, G1, S, and G2/M) was analyzed using FlowJo v.10 Software (FlowJo LLC, OR, USA).

#### 2.9.4. Clonogenic Assay

Cells were daily treated with EVs, and, after 72 h, cells were trypsinized, seeded in 6-well plates (20 cells/cm^2^), and incubated at 37 °C and 5% CO_2_ for 10 to 15 days, with EVs and/or drugs-free media. After forming visible clones, cells were washed with PBS twice, fixed with formaldehyde (3.7%) for 10 min, and stained with crystal violet (0.5%) for 15 min. Clones with >30 cells were counted under the microscope.

#### 2.9.5. BRAF and MEK Inhibitors Treatment

Cells were seeded in 24-well plates (10^4^ cells/well). Twenty-four hours later, BRAF (PLX-4032; Selleckchem) and MEK (PD-325901; Sigma-Aldrich) inhibitors were added (1:1 µM) to cell media. After 72 h, cell number, cell cycle profile, and clonogenic assays were conducted. We used DMSO (0.1% *v*/*v*) dissolved in PBS as a vehicle for the injection of MAPKi.

### 2.10. Electroporation of EVs

For intratumoral injection, EVs were exogenously loaded with miR-195-5p or control mimic (10 nM). The protocol to electroporate EVs was adapted from Pomatto et al. [16]. First, EVs were obtained from conditioned media of untreated melanoma cells by ultracentrifugation and quantified using NanoSight NS300, as described above. Then, 10^9^ EVs were resuspended in 150 μL of electroporation buffer (citric acid 0.02 M, disodium phosphate 0.03 M, and EDTA 0.1 mM, pH = 4.4) and electroporated using 10 pulses of 10 ms and 750 V in BTX equipment (Gemini). Next, the suspension was incubated at room temperature for 15 min to allow EVs membrane reintegration, ultracentrifuged at 100,000× *g* for 2 h (4 °C), resuspended in 100 μL of PBS 1X, treated with RNAse A, and quantified by NTA, as previously described. MiR-195-5p loading was confirmed by RT-qPCR after EVs treatment with RNAse A.

### 2.11. In Vivo Tumor Growth

Mice (male, Balb/C nu/nu, 9-week-old) were inoculated subcutaneously with A375 cells plus electroporated EVs. As described above, cells were previously treated in vitro with EVs or EVs plus MAPKi. Then, after 72 h, cells were detached with trypsin solution and counted using a Newbauer chamber under microscopy. Cells were resuspended in RPMI media and admixed with eEVs, and 10^6^ cells were s.c inoculated in the ventral region of each animal. Intratumorally EVs injections (10^8^ particles) were realized every 3 days after tumors reached palpable size. Tumor growth was monitored daily, and tumors were measured using a caliper. Tumor volume was calculated using the formula V (volume) = larger diameter × (smaller diameter)^2^/0.52. At the end of the experiment, animals were euthanized, and the tumors were removed for gene expression analysis. Animals were kept under a 12:12 light-dark cycle with water and food ad libitum. All in vivo experiments were conducted according to legislation for animal research in Brazil with approval from the Committee on Ethics of Animal Experiments of the University of São Paulo (approval number 1637/2021).

### 2.12. In Silico Analysis

miRNA target prediction analyses were conducted using TargetScan Human v8.0 (Appendix A). Gene expression levels from The Cancer Genome Atlas (SKCM/20160128) were downloaded using UCSC Xena Browser 2.0. Pearson correlation test was used to determine the correlation between miR-195-5p and CCND1, YAP1, or BCL2-L1 mRNA levels.

### 2.13. Statistical Analysis

Differences between three or more groups were analyzed by one or two-way ANOVA with Bonferroni post-test and between two groups by one-way ANOVA with t Student post-test, using GraphPad Prism Software v8.01. *p* ≤ 0.05 was considered statistically significant.

## 3. Results

### 3.1. MiR-195-5p Overexpression Modulates Melanoma Cells’ Vesiculation Profile

To address whether miR-195-5p overexpression could influence EVs secretion, A375, SKMel-5, and SKMel-28 human melanoma cells were transfected with miR-195-5p or negative control mimic for 72 h. Transfection efficiency was confirmed by RT-qPCR, showing increased miR-195-5p levels in all cell lines (Appendix A). In accordance with results previously reported by our group [12], miR-195-5p overexpression exerted an anti-proliferative effect in melanoma cells which was characterized by a reduction in the percentage of cells in G2/M and an increase in hypodiploid cells (Appendix A; *p* < 0.01).

Next, nanoparticle tracking analysis (NTA) showed that cells overexpressing miR-195-5p released more EVs compared to the control (Figure 1a,b, *p* ≤ 0.001). In both groups, EVs presented diameters compatible with small vesicles, ranging from 150 to 220 nm. A375 and SKMel-28 cells overexpressing miR-195-5p released smaller EVs compared to the control (156 nm compared to 188 nm for A375 and 197 nm compared to 219 nm for SKMel-28) (Figure 1c, *p* ≤ 0.01). As expected, TEM images revealed round-shaped EVs with a similar diameter range to those observed by NTA (Figure 1d). Additionally, Western blot analysis confirmed the presence of classical EVs markers (CD9 and CD63) and the absence of calnexin in the EVs pellet (Figure 1e), showing that EVs were not contaminated by cell debris.

To gain insight into the molecular mechanism behind increased EV secretion, we analyzed the expression of genes involved in EVs’ biogenesis. Upon miR-195-5p overexpression, we observed a higher expression of GTPases involved in exosomes biogenesis, including RAB27B, RAB31, and FLOT2, and decreased expression of CDC42 and RHOA, both involved in microvesicle (large vesicle) shedding (Figure 1f). Importantly, transfection of negative control mimic did not affect the cell vesiculation profile compared to cells incubated with lipofectamine without exogenous mimics (Appendix A), evincing that miR-195-5p can indeed induce the secretion of small EVs.

### 3.2. Overexpressing miR-195-5p Alters EV miRNA Content

In addition to the effect on EV production and secretion, we investigated whether miR-195-5p could also modify EVs cargo, focusing mainly on microRNA content due to their oncogenic or suppressive role in human cancers [17]. We detected 182 and 20 miRNAs in melanoma cells and derived EVs, respectively. Among these, 102 miRNAs were common to the three cell lines, while 14 were found in the secreted EVs (Figure 2a). Hierarchical clustering based on the levels of common miRNAs for cells and EVs showed that cells could be grouped by stage, independently of miR-195-5p expression status, with the primary melanoma cell line (A375) presenting a different expression pattern compared to the metastatic ones (SKMel-5 and SKMel-28). On the other hand, EVs can be distinguished according to miR-195-5p expression levels in their parental cells, independently of the cell line, indicating that miR-195-5p levels in melanoma cells modulate EVs miRNA content (Figure 2b).

Upon miR-195-5p overexpression, 41 miRNAs were found differentially expressed (fold change <0.5 or >2) in melanoma cells; 5 of these miRNAs were commonly up or down-regulated in all cell lines (Figure 2c). As expected, miR-195-5p presented the highest enrichment, followed by the tumor suppressors miRNAs miR-202-3p, miR-152-3p, and miR-424-5p, while miR-16-5p was down-regulated (Figure 2d). For the EVs, 13 miRNAs were found differentially abundant (fold change <0.5 or >2), with three out of the four miRNAs up-regulated in the cells also presenting increased abundance in all EVs (Figure 2d), indicating that re-expression of miR-195-5p modifies microRNA expression profile in melanoma cells and derived EVs, leading to an increase in tumor suppressors microRNAs. Although miR-195-5p overexpression significantly enriched miR-152-3p and miR-202-3p within EVs, miR-195-5p represents more than 90% of the transcripts present in these vesicles (Table 1). We then focused on the effect of this particular miRNA inside tumor vesicles for further experiments.

### 3.3. MiR-195 EVs Exerts an Anti-Proliferative Effect in Bystander Tumor Cells

We next asked whether EVs secreted by miR-195-5p overexpressing melanoma cells (named miR-195 EVs) affect bystander cells. Fluorescence microscopy images obtained after overnight incubation with PKH26 labeled EVs revealed a perinuclear and cytoplasmic incorporation profile (Figure 3a), indicating that EVs are used as a communication route in melanomas. Next, as shown in Figure 3b, tumor bystander cells (i.e., non-overexpressing miR-195) were treated with different EVs concentrations for 72 h when we observed that miR-195 EVs were able to affect cells’ viability (*p* < 0.001) (10^8^ EVs/mL for A375, 5 *×* 10^6^ and 10^7^ EVs/mL for SKMel-5 and 5 *×* 10^7^ and 10^8^ EVs/mL for SKMel-28 cells. Based on these data, we chose 10^8^, 10^7^, and 10^8^ EVs/mL as the optimum concentrations for the following experiments for A375, SKMel-5, and SKMel-28, respectively. Concerning cell proliferation, miR-195 EVs also induced a cytostatic effect in these recipient cells, resulting in a lower number of viable cells (Figure 3c, *p* ≤ 0.01).

To further analyze the mechanisms associated with this anti-proliferative effect, we conducted gene expression analysis using a panel of genes involved in cell cycle regulation (Appendix A). Our results showed that miR-195 EVs treatment correlated with decreased expression of CCND1 and YAP1, both genes involved in the induction of cell cycle progression (Figure 3d). These findings were validated in melanoma cancer public datasets (TCGA SKCM/20160128), where a negative correlation between miR-195-5p and CCDN1 or YAP1 expression levels was observed (Figure 3e). Interestingly, target enrichment analysis showed that CCND1 is a common target for miR-195-5p, miR-152-3p, and miR-202-3p (Figure 3f), while YAP1 is a predicted target of miR-195-5p (Appendix A). Since we had detected that miR-195-5p is packaged into small EVs, we reasoned that the reduced levels of CCND1 and YAP1 mRNA in bystander cells are due to an increase in miR-195-5p through EVs transfer. To confirm this finding, we examined miR-195-5p levels in recipient cells and observed an increase in miR-195-5p expression after miR-195 EVs treatment, confirming it was functionally transferred to bystander cells (Figure 3g). Importantly, treatment of EVs with RNAse A did not impair their effect, and EV-depleted CM had no impact in recipient cells, demonstrating that the cytostatic effect is exerted through cargo transfer by EVs (Appendix A).

### 3.4. MiR-195 EVs Sensitize Recipient Cells to Combined Targeted Therapy

Since miR-195-5p sensitizes melanoma to therapy, as our group previously showed, we sought to investigate whether the secreted EVs could also imprint this effect in recipient cells. Then, melanoma cells were treated with BRAF and MEK inhibitors (Appendix A) plus EVs (daily added) for 72 h. Interestingly, we observed that treatment with miR-195 EVs alone could impact the clonogenic potential of SKMel-5 and SKMel-28 cells (Figure 4c, *p* < 0.01), showing that EVs cargo transfer could exert a sustained effect in recipient cells’ phenotype. Moreover, upon MAPKi treatment, the addition of miR-195 EVs increased therapy response, resulting in a lower number of viable cells (Figure 4a) (*p* < 0.05), increased percentage of cell death (Figure 4b) (*p* < 0.05) and decreased clonogenic capacity (Figure 4c) (*p* < 0.05) for all the cell lines.

MAPK inhibitors are known to induce up-regulation of some stemness-related genes and anti-apoptotic and metalloproteinases genes [18]. Based on this, we analyzed gene expression using a panel of transcripts associated with cell proliferation, survival/death, stemness, and EMT (Appendix A). In accordance with this, we also noticed increased expression of Yamanaka genes SOX2, NANOG, and POU5F1, besides the metalloproteinase and anti-apoptotic genes MMP1 and BCL2-L1, respectively, after MAPKi treatment. However, when these cells were treated with MAPKi in the presence of miR-195 EVs, the gene expression profile imprinted by MAPKi was modified. We detected a significant down-regulation in BCL2-L1 -l1 levels in all 3 cell lines (Figure 5a,b). Interestingly, target prediction analysis showed that BCL2-L1 is a predicted target for miR-195-5p and miR-202-3p (Appendix A) and, to confirm that the axis miR-195-5p/Bcl2-L1 is involved in the sensitization of melanoma cells through EVs, we evaluated BCL2-L1 protein levels and observed a significant decrease in SKMel-5 and SKMel-28 cells (Figure 5c). Accordingly, TCGA data showed a significant negative correlation between miR-195-5p and BCL2-L1 expression in melanoma (Figure 5d).

### 3.5. EVs Exogenously Loaded with miR-195-5p Impairs In Vivo Melanoma Growth and Repopulation Capacity

EVs have been proposed as a promising tool for specific drug delivery, and, concerning RNAi therapy, it was shown that small vesicles (exosomes) exhibit a superior ability to deliver RNAi compared to liposomes and suppress tumor growth in a pre-clinical model [19]. Therefore, here we explored the therapeutic potential of miR-195-5p delivery by EVs. We exogenously loaded melanoma-derived vesicles with this miRNA and analyzed their effect in a xenograft tumor model. Notably, the electroporation process did not induce significant effects in EVs morphology as evaluated by NTA (Figure 6a), and RT-qPCR confirmed the enrichment of miR-195-5p within electroporated EVs (eEVs) (Figure 6b). It is worth mentioning that EVs were treated with RNAse A after electroporation to eliminate possible contamination by the presence of miR-195 outside of vesicles. After in vitro treatment, cells admixed with eEVs were subcutaneously inoculated in Balb/c nude mice. Once tumors were palpable, eEVs intratumorally injections were initiated and conducted every 3 days until the end of the experiment (Figure 6c). As shown in Figure 6d, 14 days after inoculation, only 40% of the mice in the miR-195 eEVs-treated group presented measurable tumors compared to 100% in the control group. Notably, one tumor completely regressed after the first miR-195 eEVs inoculation. Concerning tumor progression, we observed a slower growth in miR-195-eEV treated group as reflected in tumor volume (Figure 6e). After 21 days, tumors were excised and evaluated for miR-195 levels to confirm their delivery by eEVs. We found higher levels of this miRNA than in control tumors (Figure 6f). Additionally, our results showed that treatment with miR-195 eEVs resulted in a significant decrease in VEGF-A expression, which we found to be a predicted target of miR-195-5p (Appendix A) and known to induce melanoma progression [20] (Figure 6f).

Finally, to examine miR-195 eEV regulation in tumor growth upon MAPKi treatment, melanoma cells were treated in vitro with MAPKi in combination with miR-195 or control eEVs and then injected s.c. in nude mice. As described above, when tumors reached a palpable size, eEVs were injected intratumorally. As shown in Figure 6g, we observed a reduced tumor formation capacity compared to cells treated with MAPKi or MAPKi plus control eEVs, indicating that miR-195-enriched vesicles can impair the repopulation capacity of melanoma cells previously exposed to drugs, resulting in a larger fraction of measurable-tumor free mice up to 56 days after injection. Once tumors reached around 500 mm^3^, mice were euthanized, and gene expression analyses were performed (Figure 6h). According to our in vitro data, bcl2-l1 expression was up-regulated in MAPKi-treated tumors compared to control and down-regulated upon miR-195 eEVs treatment (Figure 6i).

Taken together, these results showed that miR-195-loaded vesicles could be engulfed by bystander tumor cells improving the efficacy of targeted therapy in BRAFV600E mutated tumors.

## 4. Discussion

Metastatic melanoma generally presents an unfavorable prognosis with poor overall and progression-free survival rates. Combined targeted treatment with MAPK inhibitors is considered the standard therapeutic approach for patients harboring BRAF mutation. Despite the significant improvements compared to response rates of conventional chemotherapy, most of these patients do not respond or relapse within a few months after treatment initiation. Here we showed that the sensitization effect of miR-195-5p in melanoma cells to MAPKi can be spread in cancer cells through the production and secretion of EVs enriched in miR-195-5p. We found that miR-195-5p overexpression in BRAF-mutated melanoma cells causes an increase in the expression levels of some EVs biogenesis-related genes like RAB27b, RAB31, and/or FLOT2, which was accompanied by a consequent increase in CD63 and CD9 positive EVs secretion. Both FLOT 2 and RAB31 are involved in forming intraluminal vesicles (ILVs), which can be released as exosomes by fusing with multivesicular bodies (MVB) with the plasma membrane; this is later regulated by RAB27b. Additionally, RAB31 also inhibits MVB degradation through the inactivation of RAB7, thus favoring the release of small EVs [21]. On the other hand, RHOA and CDC42 were down-regulated in A375 and SKMel-28 cells after miR-195-5p overexpression, arguing that miR-195-5p could impair the release of larger EVs since RHOA and CDC42 are involved in microvesicle shedding [22].

According to knowledge that EVs cargo reflects the phenotypic state of their parental cell, we noticed that under restoration of miR-195-5p expression in melanoma, EVs microRNA cargo was also altered. By analyzing the miRNA profile within EVs and their parental cells, we found that overexpressing miR-195-5p results in increased levels of its own as well as miR-152-3p and miR-202-3p levels in cells and derived vesicles. As previously shown by our group, miR-195-5p is among the miRNAs controlling melanoma growth and therapy response [12]. It belongs to the miR-15/16/424/497 family of tumor suppressor miRNAs, known to exert antitumoral effects in different tumor types, including lung, colon, prostate, and breast cancer [23,24,25,26]. In the same way, miR-152-3p and miR-202-3p have been described as tumor suppressors in several cancers [27,28,29,30,31,32,33,34]. Although the precise mechanism is still unknown, our findings suggest that miR-195-5p is involved in loading specific tumor suppressor microRNAs into extracellular vesicles.

Since EVs are important mediators in tumor progression, we further investigated the effect of miR-195-enriched EVs in bystander melanoma cells, and we found that these vesicles can impair recipient cells’ growth. Li et al. [35] obtained similar results by overexpressing miR-195 in fibroblasts, which resulted in the release of EVs enriched with this miRNA, inhibiting tumor growth and invasiveness in the cholangiocarcinoma model. At the molecular level, we demonstrated that miR-195-5p is effectively transferred to bystander cells, leading to a decrease in CCND1 and YAP1 expression levels which are predicted targets of miR-195-5p. CCND1 is usually up-regulated in melanomas and associated with primary tumor growth and expansion [36]. Similarly, YAP1 was recently shown to correlate with melanoma proliferation and resistance to MAPK inhibitors, and it is considered a potential target to overcome therapy resistance [37]. Then, our results suggest that the cytostatic effect caused by miR-195 EVs delivery might be caused by a reduction in CCND1 and YAP1 levels. Indeed, we observed that miR-195-5p expression level is negatively correlated with both genes in melanoma patients, reinforcing the finding that miR-195-5p/CCND1 and miR-195-5p/YAP1 axis are involved in human melanoma progression.

Recently, the role of EVs in the modulation of treatment response has gained significant attention. Specific miRNAs within melanoma EVs have been associated with patients’ prognosis and response to targeted treatment [8,11,38]. Under this scenario, we investigated the effect of miR-195 EVs in drug-targeted therapy with MAPKi. We did observe an improvement in MAPKi efficacy in the presence of miR-195 EVs characterized by a decrease in clonogenic capacity and an increase in cell death ratio. Interestingly, the expression level of some genes known to be important modulators of stemness state, proliferation, and cell death signaling was altered by MAPKi and miR-195 EVs in melanoma cells. In particular, the combined treatment with miR-195 EVs and MAPKi decreased the anti-apoptotic molecule BCL2-L1 expression at both mRNA and protein levels. In accordance with these findings, our in silico analysis showed that BCL2-L1 is a predicted target of miR-195-5p and miR-202-3p, and the TCGA data showed a negative correlation between this gene and miR-195-5p in SKCM. BCL2-L1 was reported to be associated with melanoma resistance [39,40]. In 2017, Trisciuglio et al. [41] showed that BCL2-L1 could induce the expression of stem cell markers in melanoma and glioblastoma cells, favoring resistant features and tumor progression. Based on these reported data and our findings, we suggest that the axis miR-195-5p/BCL2-L1 is involved in melanoma sensitivity to targeted therapy which can be explored to avoid tumor repopulation after MAPKi treatment.

In addition to the innate capacity of EVs to efficiently transfer active cargo between cells, there is considerable interest in the potential of enriching tumor suppressor miRNAs within EVs for cancer therapeutics. Thus, to explore the translational potential of delivering tumor suppressor miRNAs by EVs, we exogenously loaded miR-195-5p within melanoma-derived vesicles for in vivo tumor treatment. In line with our in vitro observations, treatment with miR-195 eEVs resulted in slower tumor growth and reduced tumor formation capacity of cells previously treated with MAPK inhibitors. Successful load of EVs enriched with tumor suppressive miRNAs into malignant melanoma cells, in vivo, demonstrates their potential as therapeutic vehicles. Indeed, endogenous EVs provide a biocompatible and immunologically safe option for delivering antitumoral molecules [42].

## 5. Conclusions

Taken together, we showed that miR-195-5p overexpression induces the secretion of EVs enriched with tumor suppressor miRNAs, which in turn can transfer antitumoral features to bystander cells in vitro, reprograming them towards a more sensitive phenotype. Exogenously loading miR-195-5p in melanoma-derived EVs also impaired in vivo tumor growth and restrained tumor regrowth after MAPK inhibitor treatment.

## Figures and Tables

**Figure 1 cells-12-01317-f001:**
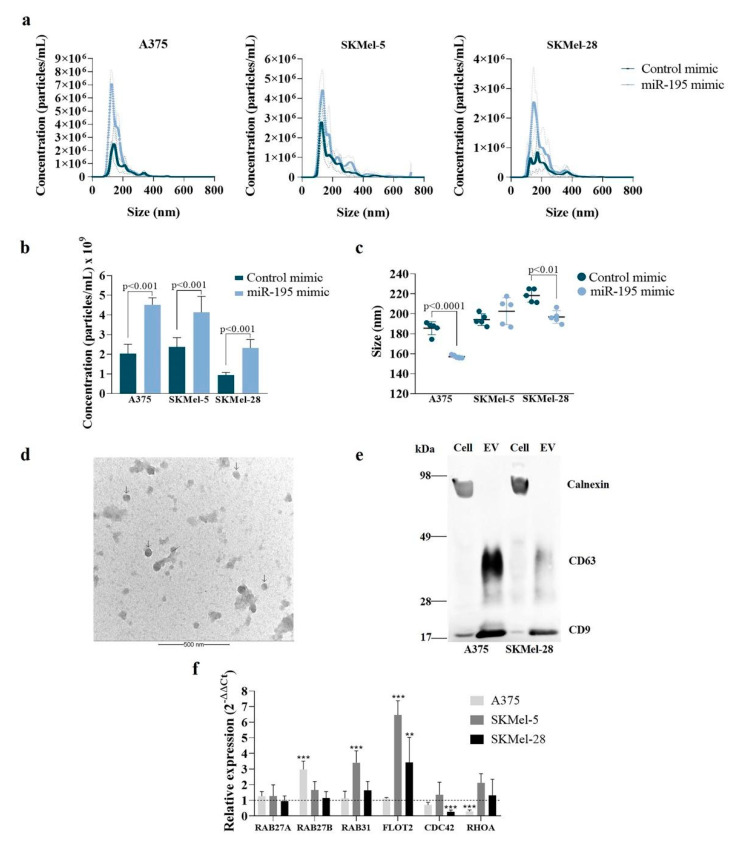
Overexpressing miR-195-5p induces small EVs release by human melanoma cells. (**a**–**c**) Cells’ vesiculation profile obtained using NanoSight NS300 (Malvem). (**a**) EVs concentration and size distribution. (**b**) EVs concentration. (**c**) EVs size. (**d**) Transmission Electron Microscopy of isolated EVs. Arrows indicates EVs. (**e**) Western blot showing classical EV markers (CD63 and CD9). Endoplasmic reticulum protein (Calnexin) was used as an exclusion marker. (**f**) Relative expression of genes involved in EVs biogenesis in miR-195-5p transfected cells compared to control. GAPDH and β-ACTIN were used as endogenous control. Statistical analysis was carried out using one or two-way ANOVA, with t-Student or Bonferroni post-test. Data are reported as means ± SD. Representative results of three independent experiments. ** *p* < 0.01, *** *p* < 0.001.

**Figure 2 cells-12-01317-f002:**
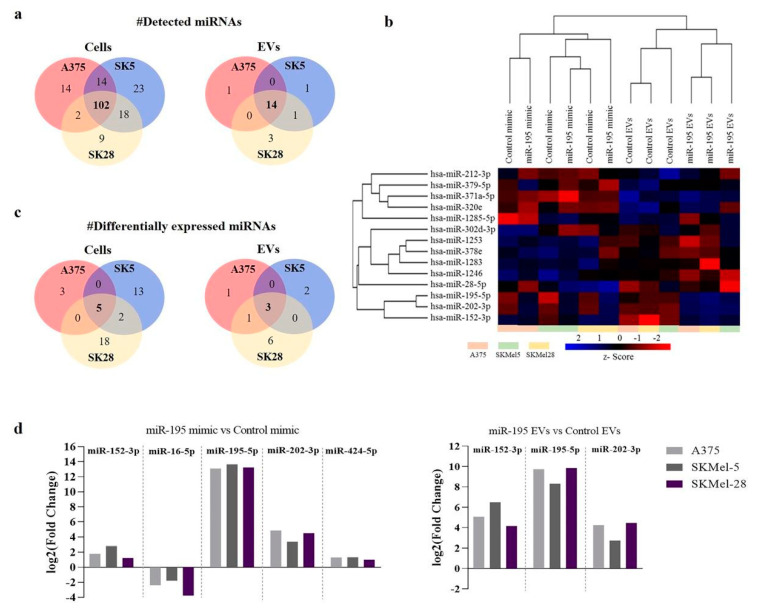
EVs released by human melanoma cells overexpressing miR-195-5p are enriched with tumor suppressor miRNAs. (**a**) Venn diagram showing the number of detected miRNAs by digital barcoding technology in cells and derived EVs from the different cell lines. (**b**) Heatmap showing the hierarchical clustering based on the levels of miRNAs commonly detected in the cells and EVs. (**c**) Venn diagram of differently abundant miRNAs (fold-change <0.5 or >2.0) in miR-195-5p transfected cells and their released EVs, compared to control. (**d**) Levels of the differently abundant miRNAs in cells and EVs.

**Figure 3 cells-12-01317-f003:**
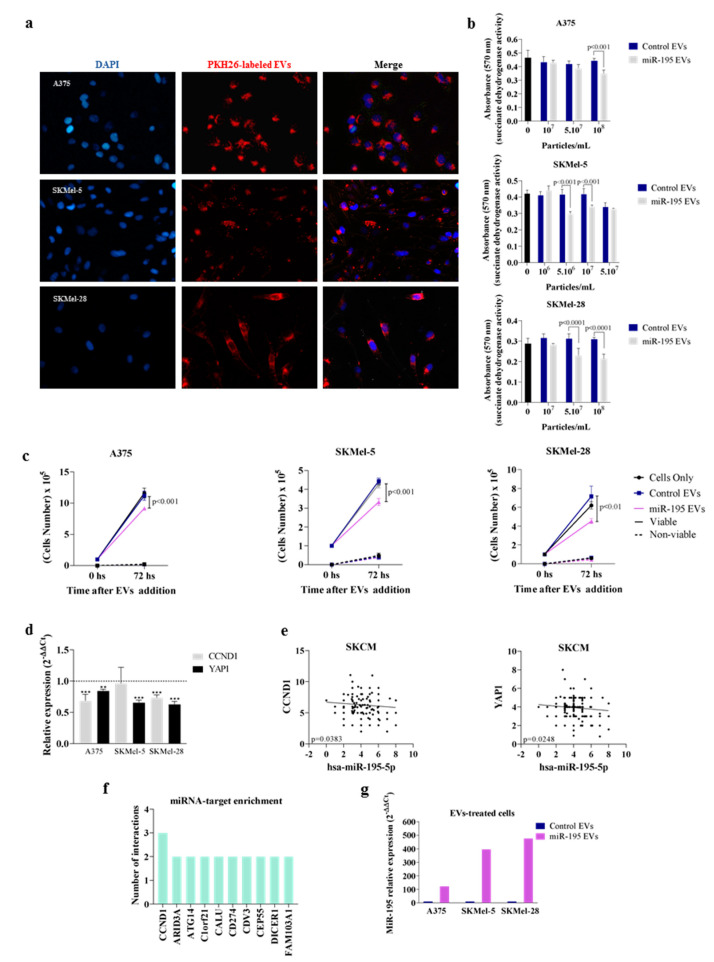
MiR−195 EVs exert a cytostatic effect in bystander recipient cells. (**a**) Incorporation profile of PKH26-labeled EVs. Magnification 10X. (**b**) Cell viability measured by MTT assay (*n* = 8). (**c**) Cell number obtained by trypan blue count (*n* = 3). (**d**) Expression level of genes involved in cell cycle regulation in miR-195 EVs-treated cells compared to control (*n* = 3). (**e**) TCGA data showing CCND1 and YAP1 correlation with miR-195-5p expression levels in SKCM. (**f**) Target-enrichment analysis showing the mRNAs predicted as common targets for miR-195-5p, miR-152-3p, and miR-202-3p. (**g**) miR-195-5p levels in recipient cells after EVs treatment. Statistical analysis was carried out using two-way ANOVA, with Bonferroni post-test. Data are reported as means ± SD. Representative results of three independent experiments.

**Figure 4 cells-12-01317-f004:**
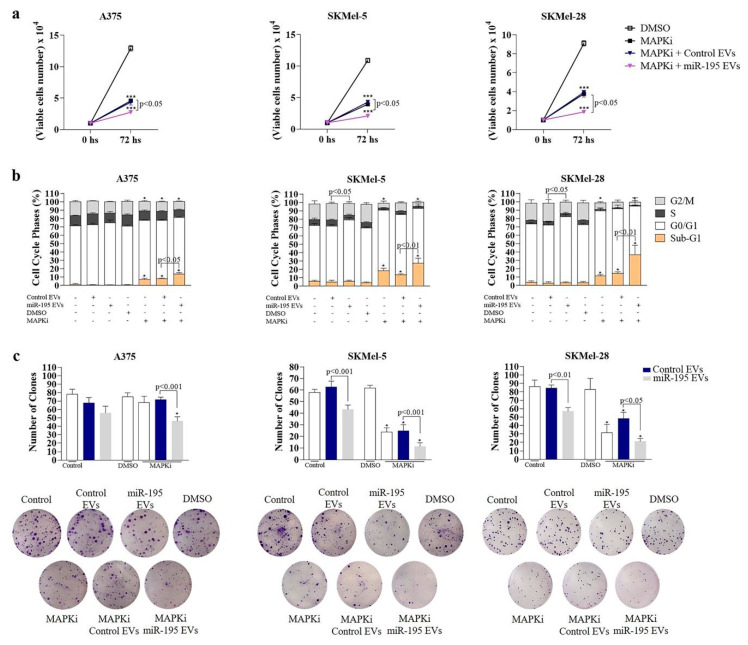
MiR−195 EVs sensitize naïve melanoma cells to BRAF and MEK inhibitors. (**a**) Cell number obtained by trypan blue count (*n* = 3). (**b**) Cell cycle profile of propidium iodide (PI)-labeled cells. (**c**) Number of clones obtained by crystal violet staining 10 to 15 days after seeding (*n* = 3). Statistical analysis was carried out using two-way ANOVA with Bonferroni post-tests. Data are reported as means ± SD. Representative results of two to three independent experiments. * *p* < 0.05, *** *p* < 0.001 (compared to control, DMSO).

**Figure 5 cells-12-01317-f005:**
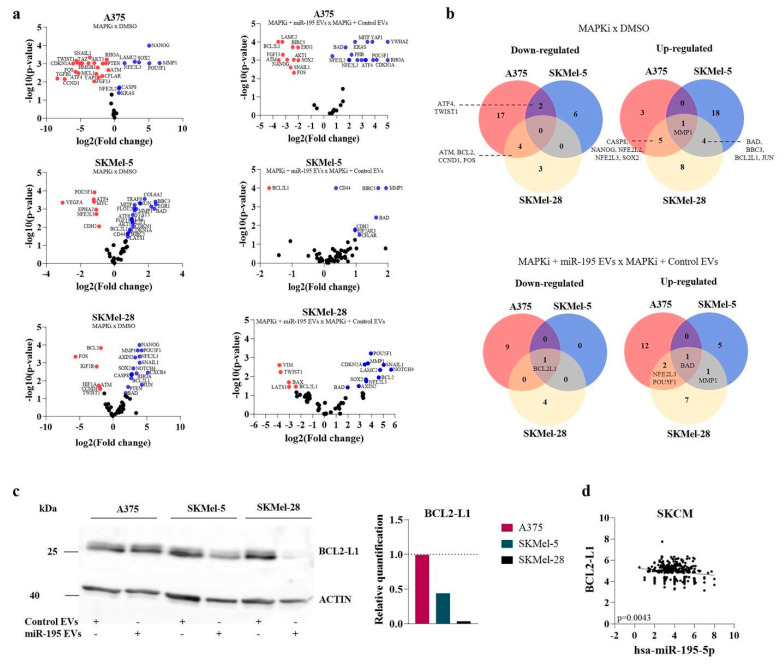
Gene expression profile after combined treatment with miR−195 EVs and MAPK inhibitors. (**a**) Volcano plot showing differentially expressed genes after MAPKi treatment (left) and MAPKi plus EVs (right). (**b**) Venn diagram representing the intersection of down and up-regulated genes in the different cell lines. (**c**) Western blot showing the protein levels of BCL2-L1 after miR-195 EVs treatment. (**d**) Correlation analysis between BCL2-L1 and miR-195-5p in SKCM from TCGA.

**Figure 6 cells-12-01317-f006:**
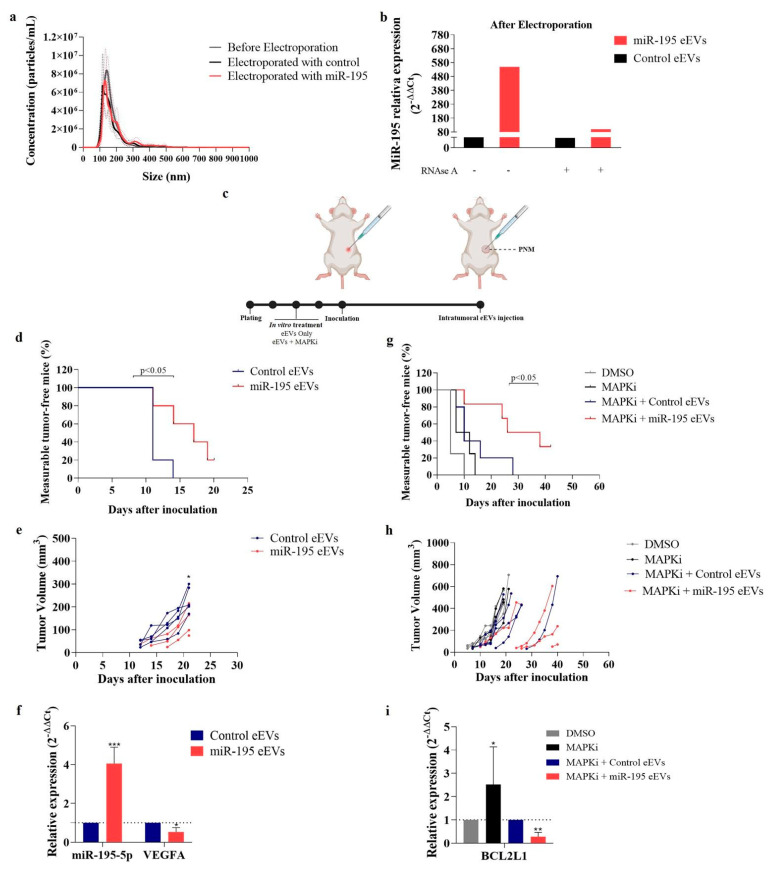
EVs carrying miR-195 impair melanoma tumor growth in vivo. (**a**) EVs’ profile before and after electroporation. (**b**) miR-195-5p levels in electroporated EVs with and without RNAse A treatment. (**c**) Experimental design. (**d**) Kaplan Meier showing the percentage of measurable-tumor-free mice treated with eEVs. (**e**) Tumor volume in the eEVs-treated group (*n* = 5). (**f**) Relative expression level of miR-195-5p and VEGA in miR-195 eEVs-treated tumors compared to control. (**g**) Kaplan Meier showing the percentage of measurable-tumor-free mice treated with eEVs and MAPKi. (**h**) Tumor volume in eEVs + MAPKi-treated group (*n* = 4 to 6). (**i**) Relative expression level of BCL2L1 in eEVs + MAPKi-treated tumors. * *p* < 0.05, ** *p* < 0.01, *** *p* < 0.001.

**Table 1 cells-12-01317-t001:** Top 10 abundant miRNAs in cells and EVs.

	Control Mimic		miR-195 Mimic		Control EVs		miR-195 EVs	
	miRNA	Normilized Count	miRNA	Normilized Count	miRNA	Normilized Count	miRNA	Normilized Count
**A375**	hsa-let-7a-5p	23,086	hsa-miR-195-5p	200,788	hsa-miR-320e	197	hsa-miR-195-5p	14,321
	hsa-miR-4454+hsa-miR-7975	10,093	hsa-let-7a-5p	21,331	hsa-miR-371a-5p	179	hsa-miR-320e	208
	hsa-miR-125b-5p	9663	hsa-miR-125b-5p	18,233	hsa-miR-1246	169	hsa-miR-371a-5p	190
	hsa-miR-29b-3p	6791	hsa-miR-4454+hsa-miR-7975	14,716	hsa-miR-1283	135	hsa-miR-630	151
	hsa-miR-100-5p	6705	hsa-miR-29a-3p	9444	hsa-miR-379-5p	105	hsa-miR-202-3p	135
	hsa-miR-23a-3p	6342	hsa-miR-100-5p	9173	hsa-miR-1253	98	hsa-miR-1283	132
	hsa-miR-15b-5p	4469	hsa-miR-29b-3p	7577	hsa-miR-765	96	hsa-miR-1246	117
	hsa-miR-19b-3p	4347	hsa-miR-23a-3p	6398	hsa-miR-378e	92	hsa-miR-152-3p	102
	hsa-miR-16-5p	3972	hsa-miR-19b-3p	5698	hsa-miR-1285-5p	86	hsa-miR-765	93
	hsa-let-7g-5p	3871	hsa-miR-15b-5p	4058	hsa-miR-630	82	hsa-miR-379-5p	91
**SKMel-5**	hsa-miR-4454+hsa-miR-7975	143,888	hsa-miR-195-5p	16,4171	hsa-miR-320e	491	hsa-miR-195-5p	14,270
	hsa-miR-29a-3p	21,561	hsa-miR-29a-3p	42,260	hsa-miR-1246	230	hsa-miR-320e	169
	hsa-miR-29b-3p	19,295	hsa-miR-4454+hsa-miR-7975	30,055	hsa-miR-1283	135	hsa-miR-1246	139
	hsa-let-7a-5p	9065	hsa-miR-29b-3p	26,661	hsa-miR-1253	126	hsa-miR-202-3p	123
	hsa-miR-21-5p	4986	hsa-let-7a-5p	13,552	hsa-miR-378e	102	hsa-miR-152-3p	110
	hsa-let-7g-5p	4632	hsa-miR-25-3p	7336	hsa-miR-379-5p	100	hsa-miR-765	100
	hsa-miR-25-3p	4464	hsa-miR-23a-3p	7071	hsa-miR-371a-5p	98	hsa-miR-379-5p	94
	hsa-miR-146a-5p	4391	hsa-miR-21-5p	6453	hsa-miR-411-5p	90	hsa-miR-1253	89
	hsa-miR-16-5p	4322	hsa-miR-15b-5p	6317	hsa-miR-765	89	hsa-miR-378e	82
	hsa-miR-15b-5p	4181	hsa-let-7g-5p	5754	hsa-miR-212-3p	84	hsa-miR-411-5p	81
**SKMel-28**	hsa-miR-29a-3p	19,593	hsa-miR-195-5p	88,671	hsa-miR-1246	174	hsa-miR-195-5p	16,913
	hsa-miR-29b-3p	18,515	hsa-miR-29b-3p	44,201	hsa-miR-765	136	hsa-miR-202-3p	178
	hsa-let-7a-5p	14,479	hsa-miR-4454+hsa-miR-7975	28,889	hsa-miR-1283	132	hsa-miR-1283	176
	hsa-miR-4454+hsa-miR-7975	13,037	hsa-let-7a-5p	23,313	hsa-miR-212-3p	119	hsa-miR-1253	155
	hsa-miR-16-5p	8450	hsa-miR-29a-3p	12,848	hsa-miR-28-5p	105	hsa-miR-1246	137
	hsa-miR-15b-5p	6679	hsa-miR-23a-3p	9675	hsa-miR-1253	100	hsa-miR-152-3p	129
	hsa-miR-23a-3p	5931	hsa-miR-15b-5p	4723	hsa-miR-320e	92	hsa-miR-378e	103
	hsa-let-7i-5p	3998	hsa-let-7g-5p	3906	hsa-miR-411-5p	89	hsa-miR-765	101
	hsa-let-7g-5p	3071	hsa-miR-25-3p	3200	hsa-miR-873-3p	86	hsa-miR-371a-5p	98
	hsa-miR-21-5p	3036	hsa-miR-374a-5p	2586	hsa-miR-302d-3p	82	hsa-miR-379-5p	84

## Data Availability

Not applicable.

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
