# Peer review of "Extracellular Vesicle-Packaged miR-195-5p Sensitizes Melanoma to Targeted Therapy with Kinase Inhibitors"

_cells, 2023, doi:10.3390/cells12091317_

Round 1

Reviewer 1 Report

In this manuscript, the authors tested whether restoration of miR-195-5p, which is downregulated in melanoma, may have therapeutic benefit in melanoma treatment. Using melanoma cell lines, they have shown that overexpression of miR-195-5p decreased cell growth, increases EV biogenesis and secretion. Additionally, miR-195-5p alters several microRNA expression profiles inside the cell as well as in secreted vesicles. Treatment of melanoma cells with miR-195-EV showed reduced growth in vitro and in vivo, sensitizes the cells to MAPK inhibitors treatment. Mechanistically, authors have proposed that the therapeutic benefit of miR-195-5p-EV is partly mediated by the downregulation of CCND1 and YAP1 mRNA. Overall, experiments were well controlled and presented clearly. In some occurrences authors overinterpreted the data which should be tested or revised. A revised version addressing the following comments will improve the clarity of the manuscript.

Specific comment:

Title: “overcomes melanoma resistance to targeted therapy with MAPK inhibitors” is an over interpretation. Combination treatment with miR-195-5p-loaded EVs and MAPKi only supports that miR-195-5p sensitizes the cells to MAPK inhibitor. To make this conclusion authors should generate a MAPK inhibitor resistance cell line and treat the resistance cells and resistant tumor with miR-195-5p-loaded EVs to test whether miR-195-5p could perturb their in vitro and in vivo growth, respectively. Otherwise, modify the title accordingly.

 Minor points:

Line 84: Is the miRNA mimic single stranded? Please clarify whether the control siRNA is a double stranded or single stranded RNA.

Line 88: Use sentence case for author’s name.

Author Response

We thank the reviewer for a careful analysis of our manuscript and positive reaction towards our findings. We agree with the reviewer about the manuscript title and we modified it as follows: Extracellular vesicle-packaged miR-195-5p sensitizes melanoma to combined targeted therapy with kinase inhibitors.

Please find below our point by point response, according to your comments.

Minor points:

Line 84: Is the miRNA mimic single stranded? Please clarify whether the control siRNA is a double stranded or single stranded RNA.

We thank the referee for having raised this issue and the opportunity to clarify it. In fact, both miRNA mimics (negative control and miR-195-5p) are double stranded molecules. We included this information in the revised version of our manuscript. 

Line 88: Use sentence case for author’s name.

We fixed this issue.

Reviewer 2 Report

- The Results should not repeat the methodology; instead, describe the methods employed in the Methodology section. 

-  Figure 3a, Table 1 , and others need editing to improve form/presentation

- Figure number and References citations, level of significance  in the same parenthesis are incorrect (e.g. order)

- There was no explanation why you used the three cell lines and this is important for those who are not familiar with those three, especially since you present the data for all three.

- Did you normalize the level expression of miR-195 in vivo (control vs treatments; vs time)? Explain how (if yes) and why not (if no)? The tumors were possibly collected at different days. 

- What vehicle was used for injection of MAPKi in combination with miR-195 or control 391 eEVs?

- There is not enough details in the in vivo part of the study. For example, the MTT assay, and other methods' descriptions are lacking. Settings, etc. 

- What is your basis for injecting 10^6 cells/animal? That seems to be too much if the cells are tumor-forming.

Author Response

We thank the referee for an in-deep revision of our manuscript. Please find below our point by point response, according to your comments.

- The Results should not repeat the methodology; instead, describe the methods employed in the Methodology section. 

We appreciate this observation and we revised the ms accordingly. All the modifications are highlighted in the revised form of the manuscript.

-  Figure 3a, Table 1 , and others need editing to improve form/presentation

We fixed these issues as suggested. 

- Figure number and References citations, level of significance in the same parenthesis are incorrect (e.g. order)

We fixed these issues.

- There was no explanation why you used the three cell lines and this is important for those who are not familiar with those three, especially since you present the data for all three.

We thank the reviewer for bringing this issue up. We used three human melanoma cell lines in order to avoid limitations including potential for inaccurate biological interpretation of data resulting from cell line-specific effects and lack of representation of melanoma heterogeneity. Then, since we observed in all tested cell lines the packing of miR-195 in melanoma derived-EVs and their effect on bystander cells, we feel that we clearly confirmed the causal link between mir-195 enriched EVs and melanoma resistance to BRAF inhibitors. 

- Did you normalize the level expression of miR-195 in vivo (control vs treatments; vs time)? Explain how (if yes) and why not (if no)? The tumors were possibly collected at different days. 

The levels of miR-195 at the in vivo experiments were analyzed when each tumor reached around 500 mm3 and were surgically excised for gene expression analysis. We chose this end time point to avoid the presence of significant necrotic areas in tumor xenografts, which might be considered a bias in results interpretation, and RNA degradation. For normalization we used as the reference sample animals injected with control eEVs, according to Livak et al (2000).

 - What vehicle was used for injection of MAPKi in combination with miR-195 or control 391 eEVs?

We used DMSO (0.1% v/v) dissolved in PBS as a vehicle for injection of MAPKi in combination with both eEVs. We added this information in the methodology section.

- There are not enough details in the in vivo part of the study. For example, the MTT assay, and other methods' descriptions are lacking. Settings, etc. 

We thank the reviewer for this comment. In the revised version of the ms, we described in greater details the in vivo study as well as the in vitro assays including all settings and controls. 

- What is your basis for injecting 10^6 cells/animal? That seems to be too much if the cells are tumor-forming.

The number of cells injected in animals were based on induction of tumor xenografts in nude mice. Data from our group concerning tumor induction from different human melanoma cell lines showed that both the engraftment and growth of this cell line in nude mice is more efficient and reproducible after injection of 106  cells in comparison to other cell quantity (105 and 5x 105, for example), providing a good model for our proposal. 

Reviewer 3 Report

* The authors investigated the effects of extracellular vesicle-packaged miR-195-5p on resistant melanoma. I found this topic interesting but language revision by a specialized company is required. 

* The title should not contain any abbreviations such as MAPK.

* Line 14: BRAF abbreviation should be defined in its first mention and all abbreviations in this abstract. 

* Line 39: BRAF abbreviation should be defined in its first mention and all abbreviations in the introduction.

* Introduce the MAPK pathway and  BRAF mutation sufficiently and their relationship to melanoma resistance. 

* Line 76: please add all catalog numbers for all used chemicals and kits.

* Line 103: how did you fix EVs on the copper grids?

* Line 132: what were the used antibodies?

* Line 190: please, mention the ethical approval number provided by your institute.

* Figure 1e: Please, repeat the western blot. 

* Figure 3a: I suggest making a quantitative analysis of the reaction area.

* Figure 6: I suggest adding Caspase-3 and Ki67

Author Response

We thank the referee for an in-deep revision of our manuscript. Please find below our point by point response, according to your comments.

* The authors investigated the effects of extracellular vesicle-packaged miR-195-5p on resistant melanoma. I found this topic interesting but language revision by a specialized company is required. 

We thank the referee for a careful evaluation of our manuscript and suggestions. Language revision was performed according to the reviewer request.

* The title should not contain any abbreviations such as MAPK.

We thank the reviewer for the suggestion and we modified it as follows: Extracellular vesicle-packaged miR-195-5p sensitizes melanoma to combined targeted therapy with kinase inhibitors.

* Line 14: BRAF abbreviation should be defined in its first mention and all abbreviations in this abstract. 

We fixed that as suggested.

* Line 39: BRAF abbreviation should be defined in its first mention and all abbreviations in the introduction.

We fixed that as suggested.

* Introduce the MAPK pathway and  BRAF mutation sufficiently and their relationship to melanoma resistance. 

This is an important point and we thank the referee for noticing this issue. We provided a more detailed background concerning the MAPK pathway and BRAF mutation in the introduction section. We added the following sentence: “Combined targeted therapy, with BRAF (B-Raf Proto-Oncogene, Serine/Threonine Kinase)  and MEK (Mitogen-Activated Protein Kinase Kinase) inhibitors, which is considered the standard first-line revolutionized treatment for BRAF-mutated patients, results in a 5-years overall survival of around 20%. BRAF mutation is the most frequent genetic alteration in melanomas, comprising around 50% of the cases. Substitution of valine with glutamate in codon 600 (V600E) is the most common mutation in BRAF gene and leads to the constitutive activation of the mitogen-activated protein kinase (MAPK) pathway, through subsequent phosphorylation of the kinases MEK and ERK, leading the the constitutive transcription of genes related to cell proliferation and drug resistance. Despite the improvements in patient’s outcomes achieved by simultaneously targeting BRAF and MEK, tumor resistance and recurrence remains a challenge, resulting in a 5-years overall survival of around 20% [1, 2].”

* Line 76: please add all catalog numbers for all used chemicals and kits.

We fixed this issue.

* Line 103: how did you fix EVs on the copper grids?

For EV TEM protocol we fixed the samples in 2% paraformaldehyde, 2% glutaraldehyde solution  (1:1) in sodium cacodylate buffer. A fixed sample of 10  µL was pipetted onto a 200 mesh copper grid, without additional fixation. Excess liquid was removed by blotting.

* Line 132: what were the used antibodies?

We fixed this issue. All the supplementary information is now included in the revised ms.

* Line 190: please, mention the ethical approval number provided by your institute.

We thank the reviewer for noticing it and included this information in the methodology section.

* Figure 1e: Please, repeat the western blot. 

Since detection of EVs markers by western blot requires a considerable amount of EVs and the generation of these nanostructures is time consuming and labor-intensive, we respectfully do not see a reason to isolate vesicles as a complementary method to demonstrate the presence of EVs and lack of cell debris in our isolates. Our decision is based on the fact that we have followed all the MISEV2018 guidelines and our blotting results are aligned with the WB data presented by several groups, characterized by not well-defined bands, like CD63, due to the presence of different post-translational modifications in this molecule.

* Figure 3a: I suggest making a quantitative analysis of the reaction area.

We thank the reviewer for this suggestion. We quantified the fluorescence  microscopy images and we did not observe any difference concerning EVs uptake among cell lines, as demonstrated below:

EVs uptake by human melanoma cells. 10 independent fields were analyzed for each condition using ImageJ software. Each cell was defined according to DAPI staining and the percentage of EVs uptake by cells were then determined by the red/blue fluorescence ratio.

* Figure 6: I suggest adding Caspase-3 and Ki67

The authors thank the reviewer for the suggestion. It would be interesting to have this additional data, however, we decided to use the xenograft tissue for RNA extraction as a way to provide a proof of concept concerning the increase in miR-195 through EVs transfer in vivo.

Round 2

Reviewer 2 Report

The Methodology part still needs improvement. There is no indication, for example where the  cells were obtained, etc. All the necessary information should be included. 

Writing can be improved by having the manuscript checked by a good editor. 

Author Response

We thank again the reviewer for the suggestions. We included additional information concerning the cell line origins and experimental procedures. In addition, the manuscript was revised by a specialized company.

Reviewer 3 Report

* The manuscript has improved significantly. However, English revision by a specialized company is necessary. 

Author Response

We thank again the reviewer for analyzing our manuscript, which was revised by a specialized company.